# Effects of Two Different Proportions of Microbial Formulations on Microbial Communities in Kitchen Waste Composting

**DOI:** 10.3390/microorganisms11102605

**Published:** 2023-10-21

**Authors:** Hairong Jiang, Yuling Zhang, Ruoqi Cui, Lianhai Ren, Minglu Zhang, Yongjing Wang

**Affiliations:** State Environmental Protection Key Laboratory of Food Chain Pollution Control, Beijing Technology and Business University, Beijing 100048, China; j2924606801@163.com (H.J.); zyyyyyl310@126.com (Y.Z.); cuiiiirq@163.com (R.C.); renlh@th.btbu.edu.cn (L.R.)

**Keywords:** composting, inoculation, harmless disposal, microbial community

## Abstract

The objective of this research was to investigate the effect of bulking agents on the maturity and gaseous emissions of composting kitchen waste. The composing experiments were carried out by selected core bacterial agents and universal bacterial agents for 20 days. The results demonstrated that the addition of core microbial agents effectively controlled the emission of typical odor-producing compounds. The addition of core and universal bacterial agents drastically reduced NH_3_ emissions by 94% and 74%, and decreased H_2_S emissions by 78% and 27%. The application of core microbial agents during composting elevated the peak temperature to 65 °C and in terms of efficient temperature evolution (>55 °C for 8 consecutive days). The organic matter degradation decreased by 65% from the initial values for core microbial agents were added, while for the other treatments the reduction was slight. Adding core microbial agents to kitchen waste produced mature compost with a higher germination index (GI) 112%, while other treatments did not fully mature and had a GI of <70%. Microbial analysis demonstrated that the core microbial agents in composting increased the relative abundances of *Weissella*, *Ignatzschineria*, and *Bacteroides*. Network and redundancy analysis (RDA) revealed that the core microbial agents enhanced the relationship between bacteria and the eight indicators (*p* < 0.01), thereby improving the bio transformation of compounds during composting. Overall, these results suggest that the careful selection of appropriate inoculation microorganisms is crucial for improved biological transformation and nutrient content composting efficacy of kitchen waste.

## 1. Introduction

In recent years, the rapid urbanization of China has led to an overwhelming amount of kitchen waste, surpassing 127 million metric tons in 2021 [1]. This surge in waste has significantly intensified the pressure on the ecological environment. Given the abundance of organic substances in kitchen waste, such as sugar, protein, oil, and lignocellulose, resource utilization has become a matter of widespread public concern [2]. One effective method to address this environmental issue is through aerobic composting, which converts kitchen residues into organic fertilizer, mitigating pollution while enhancing the value of kitchen waste [3]. During this composting process, microorganisms facilitate the transformation of organic waste into stable and humic substances, generating heat, carbon dioxide, ammonia, and water [4]. The resulting compost product can then be utilized as fertilizer or soil amendments. Despite the benefits of composting, many developing countries, including China, still rely on traditional composting due to its simplicity and affordability [5]. However, this approach has numerous drawbacks, including extended fermentation cycles, significant odor pollution, limited harm reduction, and poor product quality. Consequently, the compost produced often fails to meet commercial standards and hinders the growth of the composting industry [6]. For successful agricultural application, compost must be stable and mature. It serves as an organic amendment to enhance plant growth, soil fertility, and carbon sequestration capacity. On the contrary, the application of unstable and immature compost can fix nitrogen in the soil, hinder plant growth by depleting oxygen in the rhizosphere, and release toxic substances [7]. Therefore, improving the efficiency of traditional composting while keeping costs low has become a crucial objective in waste management for developing countries.

In this context, the inoculation of beneficial microorganisms emerges as a promising strategy to promote the composting process. Such inoculation can yield several advantages, including reducing the initial lag phase, stimulating the production of various enzymes, expediting the decomposition of organic wastes, enhancing humification, improving the quality of the final product, and ultimately reducing production costs [8]. Abdel-Rahman et al. [9] utilized pure and mixed cultures of cellulolytic bacteria to accelerate the composting of a mixture containing rice straw and cow manure. Similarly, Kausar et al. [10] employed commercial microbial inoculants (EM and LDD1) to facilitate the degradation of kitchen preparation scraps mixed with dry leaves. Wang et al. [11] took a different approach, combining twenty-five different cellulosic isolates from chicken manure and maize straw compost with seven purchased strains. The utilization of this complex inoculum had positive effects on the thermophilic phase, elevating the temperature, and producing a mature product with a higher GI (96% compared to 62%). In another study, Xu et al. [12] introduced *Bacillus licheniformis*, *Aspergillus nidulans*, and *Aspergillus oryzae* as a mixed microbial inoculant during the early phases of co-composting dairy manure and sugarcane leaves. The inoculated pile displayed a rapid succession of microbial populations involved in the decomposition and transformation of organic matter and lignocellulose [13]. Moreover, the average ammonia emissions from the inoculated composts ranged between 109 and 202 mg/m^3^ during 2~20 days of the composting process. The observed NH_3_-N losses in all treatments accounted for 15~23% of the total N loss. Previous research has also demonstrated that the exogenous addition of bacterial agents is more convenient and economical [14,15]. Sarkar et al. [16] reported that inoculating *Geobacillus* during the thermophilic stage significantly improved microbial metabolism in composting and accelerated the composting process. However, it should be noted that some studies have indicated that the introduction of inoculated bacteria might compete with indigenous beneficial microorganisms, potentially leading to suboptimal composting effects [17].

In recent years, there has been a growing interest in optimizing composting processes, particularly through the incorporation of microbial inoculums [18]. Composting with microbial inoculums offers numerous advantages, as it enhances the composting process and improves the metabolic capacity of core bacteria. Additionally, the introduction of biochar from various raw materials during composting has been shown to increase the relative abundance of bacterial diversity [19]. However, existing studies have not clearly elucidated the physicochemical parameters that impact the potential bacterial communities during composting.

The objective of this study is to investigate the influence of microorganisms on the variation in bacterial community assembly during a 20-day kitchen waste composting process using two different inoculants, as determined by high-throughput sequencing. The physicochemical properties, NH_3_, and H_2_S emissions were regularly monitored throughout the kitchen waste composting process. The correlation between the bacterial community and environmental factors in the two different inoculants was assessed through network analysis and redundancy analyses. The results of this study are focused on exploring the role of core microorganisms in composting, providing further research for the development of microbial agents for efficient deodorization and improving the composting process.

## 2. Materials and Methods

### 2.1. Composting Experiment and Sampling

Kitchen waste from the canteen of Beijing Technology and Business University served as the raw material for aerobic composting. The raw materials consisted of kitchen waste and wheat straws, mixed at a ratio of 10:1 (*w*/*w*). The experiment was conducted in three parallel cylindrical fermentors with dimensions of 1.2 m in height and 0.8 m in width for 20 days (Figure 1A). Each cylindrical fermentor received approximately 150 kg of fresh kitchen waste and 15 kg of wheat straw mixture. The initial moisture content of the composts was around 70%. The external temperature during the composting process was around 18 °C. The initial salt content of kitchen waste was less than 1%, and the DO of the reactor was between 15~20%. The initial pH value of fresh kitchen waste was 7. The microbial inoculants, designated C1 and C2, were provided by the Ecological Environment Laboratory of Beijing Technology and Business University. C2 contained six cultures representing Firmicutes, Bacteroidota, Actinobacteriota, Proteobacteria, Deinococota, and Chloroflexi. On the other hand, C1 had four additional cultures representing Halanaerobiaeota, Cyanobacteria, Desulfobacterota, and Halobacterota. On days 1, 8, and 15 of composting, we added 3 L of C1 and C2 microbial inoculum to the corresponding cylindrical fermentor. The composting process involved static composting with frequent turning for the blank treatment (CK). Mechanical turning was carried out daily from day 0 to day 20. Representative triplicate samples of each compost (~100 g) were collected on different days (2nd, 4th, 6th, 8th, 10th, 12th, 14th, 16th, 18th, and 20th) and were stored at 4 °C for physical-chemical analysis. Additionally, samples from the 1st day, 10th day, and 20th day (~100 g each) were preserved at −20 °C for DNA extraction to study the microbial content.

### 2.2. Composting Analytical Methods

NH_3_ and H_2_S concentrations in the ambient air were directly measured using a sensor (S-NHHS, China). The sampling duration was 1 h to determine the average concentration over that period. Temperature was monitored using temperature sensors (HIH-3602, China). The gas measurement points and temperature measurement points are shown in Figure 1B. The pH of the compost was measured in a water suspension (1:10, *w*/*v*) with a pH meter (BPH-7100A, China). The E4/E6 ratio was measured using an ultraviolet spectrophotometer at wavelengths 465 nm and 665 nm (UV-5100H, China), which characterizes the maturity index of composts [20]. Fresh composting samples were oven-dried at 105 °C (for approximately 8 h) to constant weight to determine the moisture content [21]. Total nitrogen (TN) and total organic matter (TOC) were measured following the Chinese National Standard (NY 525-2012) [22]. The seed germination index was calculated using the Chinese National Standard (NY 525-2021) [23]. All samples were tested in triplicate.

Total genomic DNA from the samples was extracted using the CTAB/SDS method. The 16S rRNA genes from the distinct regions 16S V3-V4 were amplified using specific primers 341F: C3TAYGGGRBGCASCAG and 806R: GGACTACNNGGGTATCTAAT with barcodes. All PCR reactions were conducted in 30 µL volumes with 15 µL of Phusion^®^ High-Fidelity PCR Master Mix (New England Biolabs, Beijing, China), 0.2 µM of forward and reverse primers, and approximately 10 ng of template DNA. The PCR products were then mixed with an equal volume of 1× loading buffer containing SYB green and subjected to electrophoresis on a 2% agarose gel for detection. The PCR products with equidensity ratios were purified using the GeneJETTM Gel Extraction Kit (Thermo Scientific, Shanghai, China). Sequencing libraries was constructed using the TruSeq^®^ DNA PCR-Free Sample Preparation Kit (Illumina, Inc., CA, USA) and sequenced on the platform of Illumina NovaSeq6000 (Illumina, Inc., CA, USA). The library quality was assessed using the Qubit@ 2.0 Fluorometer (Thermo Scientific, Shanghai, China). Finally, the paired-end raw sequencing data can be obtained through NCBI under accession number SRR26373766, SRR26373767, SRR26373768, SRR26373769, SRR26373770, SRR26373771, SRR26373772, SRR26373773 and SRR26373774. The raw sequences were analyzed using FASTP (v0.18.0, https://github.com/OpenGene/fastp) (accessed on 14 February 2022) for quality control to get high-quality clean reads. Paired-end clean reads were merged as raw tags using FLASH2 (v2.2.00, https://github.com/dstreett/FLASH2) (accessed on 14 February 2022) with a minimum overlap of 10 bp and mismatch error rates of 2%. The reads were replicated, sorted, and clustered into operational taxonomic units (OTUs) at the default 97% similarity using UPARSE (v7.0.1001, http://drive5.com/uparse/) (accessed on 14 February 2022) [24], and a representative sequence for each OTU was selected for further annotation.

### 2.3. Statistical Analysis

Alpha-diversity, including Chao1, Shannon and Simpson diversity index (H), explained the richness and diversity of bacterial community in all samples using vegan package in R [25]. To evaluate the differences in microorganism communities among all treatments, Venn, non-metric multidimensional scaling analysis (NMDS), Principal Component Analysis (PCA) and Principal Coordinates Analysis based on Bray-Curtis distances (PCoA) were conducted. The Draw Venn Diagram was used for Venn. The NMDS, PCoA and PCA were performed following the procedure outlined by Ghanizadeh and James [26]. Redundancy Analysis (RDA) was employed to assess the correlation between microorganism communities and the physicochemical properties of the composts. For the RDA, the vegan package’s rda function was utilized for the sorting analysis. Molecular Network Analysis used a cutoff level of 0.95 to enable comparison between composting treatments. The qgraph package was used for Network Analysis. The threshold values (Zi and Pi) were used to calculate each node’s topological functions [10].

## 3. Results

### 3.1. Changes in Physicochemical Characteristics

Figure 2 illustrates the changes in temperature (T), pH, E4/E6 ratio, moisture (MC), total nitrogen (TN), and total organic carbon (TOC) during composting in three treatments. Temperature is a crucial factor influencing organic matter decomposition and microbial growth during the composting process. The composts with microbial inoculation (C1 and C2) exhibited higher temperatures than the CK (control) during composting, indicating the significant role of exogenous inoculum in rapid pile warming (Figure 2A). Furthermore, C2 showed a longer duration of the high-temperature phase (>55 °C) than C1. Notably, in the initial stage of the composting experiment, the C2 treatment rapidly reached a temperature of 65 °C on day 7. The changes in pH during all composting processes followed a similar pattern (Figure 2B). The initial decrease in pH was linked to the production and accumulation of organic acids through the decomposition of easily degradable organic substances. However, the final pH of CK and C1 was 6.09 and 5.60, respectively, lower than C2.

Throughout the composting process, the E4/E6 ratio for all treatments showed a decreasing trend (Figure 2C). The initial moisture content for all treatments was approximately 70%, and then it tended to decrease as the process progressed (Figure 2D). The most substantial decrease in moisture content was observed in C2 (34%), while CK and C1 experienced reductions of only 18% and 24%, respectively. The changes in TN content showed a different trend among all treatments (Figure 2E). There was a noticeable decrease in TN during composting in C2 compared to CK and C1. Additionally, the TOC content decreased during composting in all treatments (Figure 2F). The removal of TOC in C2 (65%) was the most substantial, whereas there was no apparent decrease in TOC in CK (conventional composting) and C1 (less than 15% removal). These findings indicated that C2 was more effective in degrading organic materials than CK and C1.

As depicted in Figure 3A, the C/N ratio gradually decreased over the composting duration, with C2 reaching a value below 15 by day 17, while CK and C1 treatments-maintained values of 20 and 18 until day 20. Statistical analysis revealed a significant difference (*p* < 0.05) in C/N values among the three treatments between 15~20 days of composting. On the last day of composting, the C/N of C2 was 13.5, 6.5 lower than CK, and 4.5 lower than C1. The growth index (GI) of all species progressively increased throughout the composting process (Figure 3B). Nevertheless, notable differences in GI values were observed among the seeds (*p* < 0.01). By the conclusion of composting, C2 exhibited a GI value of 112%, significantly higher than CK and C1 treatments (49% and 68%). In this study, the temperature and physicochemical indices indicated that the final compost of C2 met the maturity requirements.

### 3.2. Variation in Emission and Cumulative Emission of NH_3_ and H_2_S during Composting

Figure 4A presented the changes in NH_3_ emissions from all treatments during 0~21 days. The NH_3_ emission in each treatment group significantly increased during the medium to high-temperature stage of composting from 6 to 15 days (Figure 2A). The cumulative NH_3_ emissions during 0~21 days were 11,000, 2854, and 673 mg/m^3^ for CK, C1, and C2, respectively. C1 exhibited a NH_3_ emission decrease of 74%, while C2 showed a larger decrease of 94%. Notably, NH_3_ emission in C2 was almost negligible. These results suggested that C2 effectively inhibits NH_3_ emissions during composting by promoting microbial utilization of NH_4_^+^.

Figure 4B displayed the changes in H_2_S emissions from all treatments during 0~21 days. The cumulative H_2_S emissions during days 0~21 were 400, 292, and 89 µg/m^3^ for CK, C1, and C2, respectively. C1 exhibited a H_2_S emission decrease of 27%, whereas C2 showed a significantly larger decrease of 78%. Compared to CK, the H_2_S emission of C1 did not change significantly, but the emission of C2 was relatively lower. Particularly, H_2_S emission was almost negligible in C2 after day 12, showing a similar trend as NH_3_ emission.

### 3.3. Spatial Distribution of Microbial Communities during Composting

#### 3.3.1. Alpha-Diversity of Microbial Community

The Venn diagram demonstrated that the number of shared operational taxonomic units (OTUs) in CK and C1 treatments initially decreased before increasing as the composting process progressed (Figure 5A–C). Compared to CK, the abundance of unique OTUs significantly increased in C1 and C2 during the late stage of composting (Figure 5B,C). Among all treatments, the abundance of unique OTUs followed the order CK > C1 > C2 at the cooling stage and C2 > C1 > CK at the mature stage, indicating that different microbial inoculants had varying effects on unique microorganisms at different phases of composting.

The Shannon rarefaction curves for observed fungal OTUs reached a saturation plateau, and the coverage index of each sample was above 0.99, indicating that the sequencing effort was sufficient to represent the entire fungal populations (Figure 5D). Basic information on sequencing, classification, and abundance of OTUs can be found in Table 1. Based on the data, the highest PD whole tree value was observed in samples from CK collected on the first day (132), while the lowest was seen in samples from C2 collected on the first day (93). Similarly, the Chao1 estimates ranged from 606 (samples from C2 collected on the first day) to 1435 (samples from CK collected on the first day). The observed species counts ranged from 995 (samples from C2 collected on the 10th day) to 1307 (samples from CK collected on the first day). The Shannon index and Simpson index provide insights into species diversity. The highest Shannon index value was observed in samples from C1 collected on the first day (6.36), indicating higher species diversity, while the lowest was seen in samples from C2 collected on the 10th day (4.68). The Simpson index ranged from 0.76 (samples from C2 collected on the 10th day) to 0.96 (samples from C1 collected on the first day), with lower values indicating higher evenness. Compared to the CK treatment, the observed species in compost samples with microbial inoculation (C1 and C2) on day 10 were significantly higher than those on day 1 (Table 1).

#### 3.3.2. Beta-Diversity of Microbial Community

The common differences and similarities among all treatments were analyzed using Beta-diversity, principal component analysis (PCA), principal coordinate analysis (PCoA), and non-metric multidimensional scaling (NMDS) (Figure 6). Figure 6A presented the dissimilarity coefficient between the two treatments, constructed using weighted and unweighted UniFrac distance in Beta-diversity analysis. The results showed differences among all treatments, with the dissimilarity coefficient increasing from the thermophilic to mature stage between 1DCK and other treatments. During the thermophilic stage, C1 and C2 treatments increased the dissimilarity coefficient compared to the CK treatment, indicating that C1 and C2 treatments improved the selectivity of bacterial communities. The largest dissimilarity coefficient was observed between samples in C1 collected on the first day and samples in C2 collected on the first day, suggesting differences in the effects of the two microbial agents on the community during the early stages of composting. According to PCA analysis, the first two principal components (PC1 and PC2) explained 34% of the total variation in microbial communities (Figure 6B). PC1 accounted for 20% of the variation and showed a positive correlation with all treatment groups, especially with samples in C1 collected on the first day and samples in C1 collected on the 10th day (Figure 6B). The PCoA analysis (Figure 6C) revealed distinct distances between the microbial communities of samples collected at the initial stage of composting and those from the later stages. PCoA1 and PCoA2 together explained 67% of the variability in the data. The results also showed that microorganism communities in the C2 treatments were clearly separated. This indicates that high temperatures have a strong screening effect on the microorganisms in the composting habitat. The NMDS analysis (Figure 6D) further confirmed that the addition of C2 at the beginning of composting led to a substantial change in the structure of the microorganism communities.

#### 3.3.3. Spatial Composition of Microbial Community

At the phylum level, Firmicutes, Proteobacteria, Bacteroidota, and Actinobacteria were the dominant phyla throughout the composting stages (Figure 7A), collectively accounting for 91~93% in CK, 84~93% in C1, and 91~97% in C2 during the composting process. In all treatments, the abundance of Firmicutes varied as composting progressed. In C2, Firmicutes increased from an initial 40% to a peak of 72% on day 10, and then gradually decreased to 42%. However, in the C2 treatment, the relative abundance of Proteobacteria decreased from 47% to the lowest value of 11% during the thermophilic phase (*p* < 0.01), and then gradually increased when the temperature decreased. This suggests that Proteobacteria could not withstand high temperatures. In contrast, the changes in Proteobacteria were not significant in CK and C1, as they did not reach high temperatures (Figure 2A). Interestingly, compared to CK and C1, the inoculation of microbial agents in C2 led to an increased relative abundance of Bacteroidota from 9% to 38%. This result explains the decrease in kitchen waste and composting height during the middle and later stages of composting. Additionally, all six strains added in C2, including Firmicutes, Bacteroidota, Actinobacteriota, Proteobacteria, Deinococota, and Chloroflexi, were present in the composting process. Deinococota showed 39 times increase, and Chloroflexi showed a 13 times increase compared to the prophase in C2. The other four bacteria added to C1 were not detected, indicating that these strains were not suitable for growth under the composting conditions.

At the genus level, the top 30 bacterial genera were identified (Figure 7B). The cluster analysis of the absolute abundance distribution of bacteria and fungi for different treatments at the genus taxonomic level was shown in Figure 7C. These changes in relative abundances were specific to each genus during the composting process in the three treatments. In the initial mixtures, *Weissella* had a low proportion in C2 but was the dominant bacterium in CK (26%) and C1 (47%). However, in the middle mixtures, *Weissella* became the dominant bacterium in C2 (55%), with abundances much higher than in CK and C1. As shown in Figure 2B, the initial pH value of C2 was 5.29, which was below the optimal growth range of *Weissella*. However, as the days of composting progressed, the pH value in C2 gradually approached the growth range of *Weissella*. In contrast, the pH values of CK and C1 did not reach the optimal growth range of *Weissella*. Therefore, in the middle stage of composting, *Weissella* became the dominant microbial community in C2. Unlike CK and C1, *Ignatzschineria* was the dominant unique bacterium in the early stage of C2, with a very low proportion in CK and C1. At the end of composting, the relative abundance of *Bacteroides* in C2 (33%) significantly increased compared to CK and C1. This increase in abundance coincided with the significant increase in pH value and the downward trend in total organic carbon (TOC) observed in Figure 2B,F, respectively. Moreover, the average relative abundance of thermophagus in C2 was higher than in CK and C1 treatments, indicating a higher prevalence of thermophilic bacteria in C2. This implies that inoculation led to an increase in the relative abundance of thermophilic bacteria, which facilitated a faster temperature rise and shortened the fermentation period.

To explore the relationship between the top ten microbial communities of the three treatments and environmental factors, RDA was conducted (Figure 7D). Eight physicochemical indices, including T, pH, E4/E6, MC, TN, TOC, C/N, and GI, were selected as the environmental factors. The physicochemical factors explained 56.57% of the variation in bacterial abundance (Figure 7D). As shown in Figure 7D, certain microbial genera, including *Ignatzschineria*, *Bacteroides*, *Keratinibaculum*, *Tissierella*, *Lysinibacillus*, and *Peptoniphilus*, were observed in the mature stage of composting, and they showed a significantly positive relationship with GI. On the other hand, *Cerasibacillus*, *Nocardiopsis*, *Pseudomonas*, and *Weissella* had a significantly positive correlation with E4/E6, MC, C/N, and TOC, respectively, and were found to be abundant in the composting process. The presence and dynamics of these bacterial communities were likely influenced by the addition of microbial agents during composting. The characteristics of physicochemical factors were essential parameters to explain the dynamics of bacterial communities at the phylum level during composting. The research on the relationship between bacterial dynamics and physicochemical factors provides valuable information for understanding and optimizing the composting process.

#### 3.3.4. Network Analysis of Bacterial Communities

In this study, the top thirty bacterial genera were selected for Pearson correlation analysis with eight indicators (T, pH, E4/E6, MC, TN, TOC, C/N, and GI) to explore their effects on the bacterial community during the three different composting treatments (Figure 8). The network analysis revealed significant differences in bacterial communities (*p* < 0.01), which could be attributed to the different inoculation of composting treatments. Each node in the network represented a bacterial genus that participated in the transformation of the eight indicators. The results showed that CK, C1, and C2 composting treatments had 34, 38, and 30 nodes, and 106, 59, and 90 edges, respectively, related to the 8 indicators transformation. In CK and C1 composting treatments, the bacterial genera showed no obvious positive or negative correlations with the eight indicators (Figure 8A,B). However, in the C2 composting treatment with the microbial agent inoculation, the bacterial community changed significantly. Most bacterial genera were positively correlated with C/N, HA, MC, TN, and TOC, while they were negatively correlated with GI, pH, and T (Figure 8C). These findings indicated that the inoculation of the microbial agent in the C2 treatment improved the relationship between bacteria and the eight indicators, thereby enhancing the biotransformation of compounds during composting.

## 4. Discussion

In the present study, the research suggested that the differences in microorganisms within the inoculants are key factors influencing the composting process [27]. Detailed analysis and comparison revealed that the inoculation with microorganisms extended the thermophilic stages from three days (C1) to eight days (C2), consistent with findings by Ajmal et al. [28]. Subsequently, a gradual increase in pH could be attributed to the degradation of organic acids and the generation of ammonia, which buffered the acidic conditions [29]. On the contrary, the decrease in pH could be attributed to the production of organic acids, a phenomenon observed in the study by Li et al. [30]. The E4/E6 ratio is a parameter used to rapidly evaluate compost maturity. It characterizes the quality of humic acid and the degree of aromatization [31]. A high E4/E6 ratio indicates a low degree of maturation, while a low ratio suggests a higher degree of maturation in the compost [32]. During composting, maintaining optimal water content is essential for microbial activities and promoting organic matter decomposition [6]. There was a noticeable decrease in TN during composting, possibly due to more significant gas emissions (e.g., NH_3_ and N_2_O) during the thermophilic stage. Compost maturity is often assessed using the C/N ratio, where a value of 10~15 is considered indicative [33]. The GI of the final compost ranged from 0 to 120%. According to the Chinese organic fertilizer standard (NY525-2021), a GI value ≥ 70% is required. NH_3_ is a significant malodorous gas produced during the degradation of organic matter in kitchen waste composting [34]. The increase of NH_3_ is attributed to the decomposition of nitrogenous materials, such as proteins and amino acids [35,36]. H_2_S is primarily produced through the synthesis pathway by sulfate-reducing bacteria in an anoxic environment [37].

The abundance of unique OTUs significantly increasing is likely attributed to the involvement of microbial inoculants in biodegradation, leading to the enrichment of bacterial communities or acting as specific microbial accumulators [38]. The richness increased, but the diversity decreased in the thermophilic phase, likely due to the propagation of thermophilic microbes. Shannon and Simpson indices are commonly used indicators for estimating bacterial diversity [39]. Higher Shannon index values indicate higher bacterial diversity, while the opposite is true for the Simpson index. This result suggests that the community structure exhibited vertical spatial differences in the compost, which aligns with similar findings reported in the composting of chicken manure [40]. The smaller values of the dissimilarity coefficient suggest homology between the two samples [41]. Additionally, the addition of microbial agents could improve the micro-environmental conditions and nutrient availability, resulting in a positive effect on bacterial communities. These findings are consistent with existing literature [30]. The clear differentiation between different phases of composting, as indicated by PCA, suggests that the structure of microorganisms undergoes significant changes as composting progresses [42]. The microorganism communities were clearly separated, implying a change in the structure of microorganism communities after the addition of microbial agents [43].

At the phylum level, Firmicutes, Proteobacteria, Bacteroidota, and Actinobacteria have also been reported as dominant in other lignocellulosic composts [44]. The increase in the relative abundance of Firmicutes was consistent with the thermophilic period of composting, as Firmicutes are known to tolerate high temperatures by forming spores, which gives them an advantage in adapting to wider ecological niches [45]. Proteobacteria was the predominant phylum in each sample, a common observation in many composting studies due to its diverse species that play important roles in the carbon, sulfur, and nitrogen cycles [46]. Zhong et al. [47] who observed a decrease in Proteobacteria during the thermophilic phase, followed by recovery in later phases. Bacteroidota is known to decompose cellulose into cellobiose and glucose, producing volatile fatty acids (VFA) in the process [48]. Actinobacteria was also abundant at a low compost height, likely due to the frequent turning promoting the growth of aerobic Actinobacteria, which thrive in higher oxygen concentrations [49]. Deinococota and Chloroflexi showed a significant increase, indicating their active participation in composting [50]. Halanaerobiaeota was negatively correlated with oxygen concentration and positively correlated with temperature, as it consists of anaerobic fermentative bacteria, favoring anaerobic fermentation [51]. Cyanobacteria were found to contribute to humus formation, but their contribution to composting was not as significant as other bacteria such as Firmicutes, Bacteroidota, Actinobacteriota, and Proteobacteria [52]. Desulfobacterota is a phylum of bacteria encompassing sulfate-reducing and related fermentative bacteria, constituting the bulk of strict anaerobes previously classified as Deltaproteobacteria [53]. The presence of Halobacterota indicates their potential to degrade cellulose and provide substrates for methanogens [54]. At the genus level, the composting process is influenced by various factors, leading to observable changes in the abundance and dominant microbial community composition [55]. *Weissella* thrives best at temperatures of 50~55 °C and pH values of 6.50~7.50. Within this temperature and pH range, *Weissella* exhibits a relatively high growth and metabolic activity, making it more effective in participating in the decomposition process of compost [56]. *Weissella* is known to promote nitrogen cycling, leading to changes in the form and distribution of nitrogen, which improves the degree of humification in compost [57]. *Ignatzschineria* is commonly found in the initial stages of composting, when the compost pile is first formed. It is believed to play a role in the initial breakdown of complex organic compounds into simpler ones, which can then be further degraded by other microorganisms [58]. Similar findings were reported by Zhan et al. [59], who also observed a decrease in the number of *Bacteroides* during composting due to an increase in pH or a decrease in TOC content. *Bacteroides* can increase the fermentation temperature and the duration of high temperatures (>50 °C) [60]. The success of composting is influenced by various environmental factors (physicochemical properties) that directly or indirectly affect microbial activity [61,62]. The microbial communities were greatly influenced by changes in these physicochemical factors [63].

The bacterial population plays a critical role in the biotransformation of organic molecules due to its biochemical diversity [64]. The microbial agent may have facilitated the cooperation between exogenous and indigenous bacteria, promoting the production of metabolites such as organic acids and enzymes during composting [65].

## 5. Conclusions

The findings of this study suggest that microbial inoculation can significantly improve the quality of kitchen waste composting. The presence of core microorganisms played a crucial role in the composting process, and their contribution was much more significant than that of non-core microorganisms. Throughout the entire composting process, the addition of only the core microbial agent in C2 treatment led to substantial reductions in NH_3_ and H_2_S emissions. The NH_3_ degradation in C2 was 78%, and the H_2_S degradation was 94%. Moreover, C2 treatment exhibited the longest thermophilic period (8 days) and a high organic matter degradation (65%) compared to the other groups. Three specific genera, namely *Weissella*, *Ignatzschineria*, and *Bacteroides*, were identified as keystone taxa that played a key role in promoting composting maturity. These genera likely synergistically contributed to the composting process. Pearson correlation analysis showed that C2 treatment had a stronger and more significant connection between functional microorganisms and the eight indicators compared to CK and C1 treatments. Taken together, these results demonstrate that suitable microbial inoculants, especially core microorganisms, are beneficial in reducing the generation of odor gases during kitchen waste composting and promoting compost maturity. Inoculation of core microorganisms is a key factor for the success of the composting process.

## Figures and Tables

**Figure 1 microorganisms-11-02605-f001:**
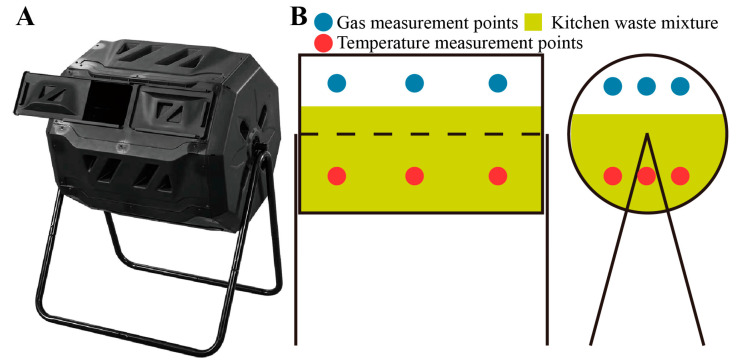
(**A**) Physical image of the cylindrical fermentor, (**B**) Gas measurement points and temperature measurement points.

**Figure 2 microorganisms-11-02605-f002:**
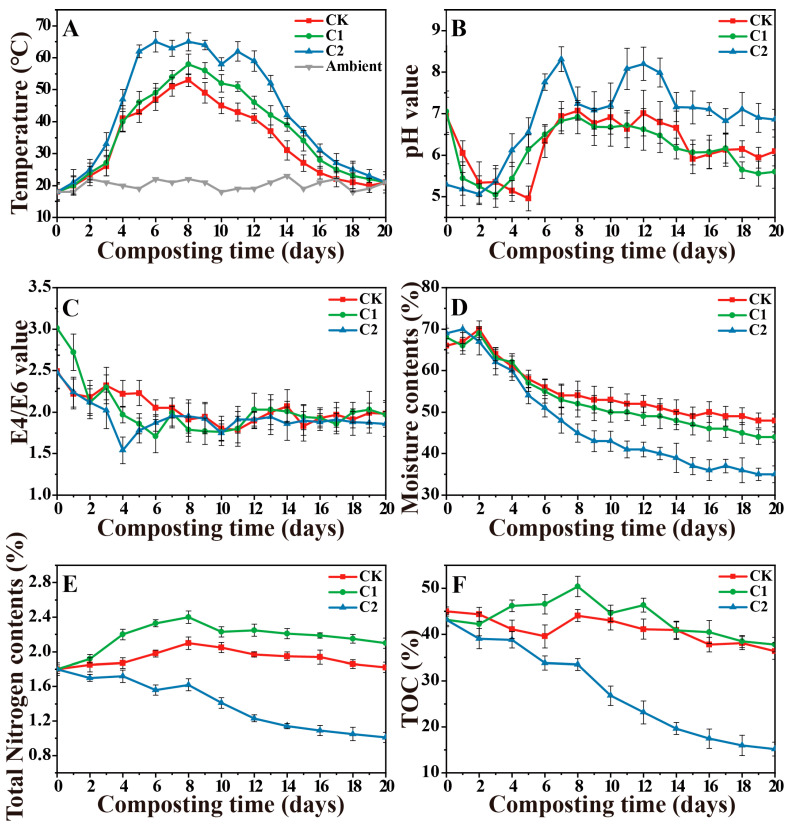
Changes of (**A**) temperature, (**B**) pH, (**C**) E4/E6, (**D**) moisture (MC), (**E**) total nitrogen (TN), and (**F**) total organic carbon (TOC) during composting in three treatments. (CK-the blank treatment was static composting with frequently turning; C1-the static composting with microbial inoculants numbered C1; C2-the static composting with microbial inoculants numbered C2).

**Figure 3 microorganisms-11-02605-f003:**
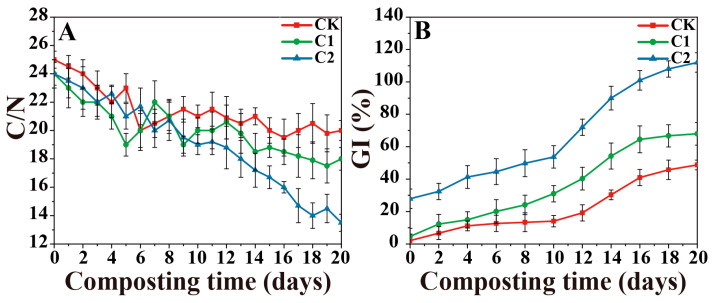
Changes in (**A**) C/N, and (**B**) seed germination index (GI) of hybrid cucumber during composting in three treatments. (CK-the blank treatment was static composting with frequently turning; C1-the static composting with microbial inoculants numbered C1; C2-the static composting with microbial inoculants numbered C2).

**Figure 4 microorganisms-11-02605-f004:**
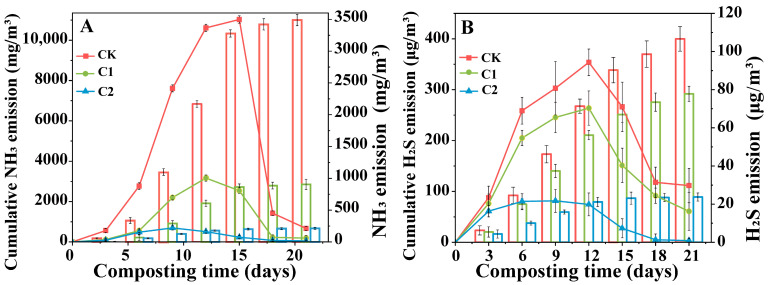
Variation in emission and cumulative emission of NH_3_ and H_2_S during composting (**A**) NH_3_, (**B**) H_2_S. (CK-the blank treatment was static composting with frequently turning; C1-the static composting with microbial inoculants numbered C1; C2-the static composting with microbial inoculants numbered C2).

**Figure 5 microorganisms-11-02605-f005:**
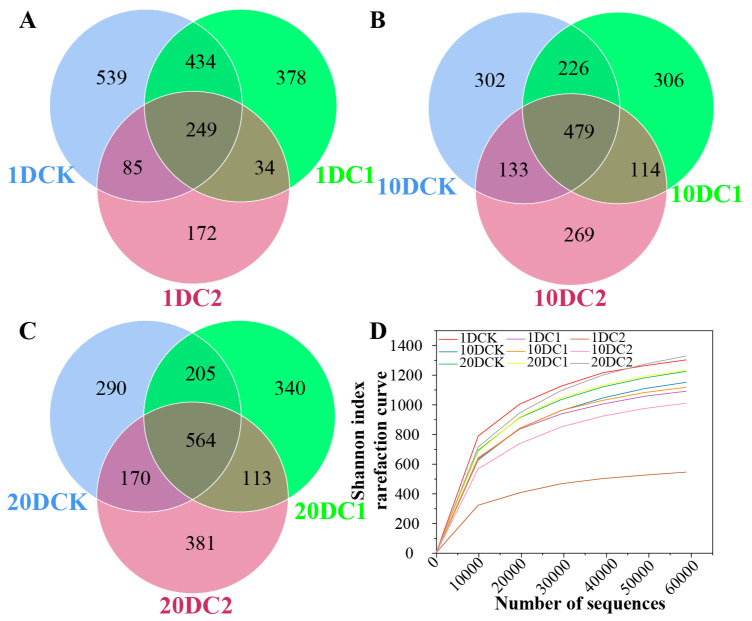
The change in Operational Taxonomic Units (OTUs) and Shannon index rarefaction curve of bacterium community structures during composting. (**A**) OTUs in 1 day, (**B**) OTUs in 10 day, (**C**) OTUs in 20 day, and (**D**) Shannon index rarefaction curve. (CK-the blank treatment was static composting with frequently turning; C1-the static composting with microbial inoculants numbered C1; C2-the static composting with microbial inoculants numbered C2; 1DCK-samples in CK collected at first day; 10DCK-samples in CK collected at 10th day; 20DCK-samples in CK collected at 20th day; 1DC1-samples in C1 collected at first day; 10DC1-samples in C1 collected at 10th day; 20DC1-samples in C1 collected at 20th day; 1DC2-samples in C2 collected at first day; 10DC2-samples in C2 collected at 10th day; 20DC2-samples in C2 collected at 20th day).

**Figure 6 microorganisms-11-02605-f006:**
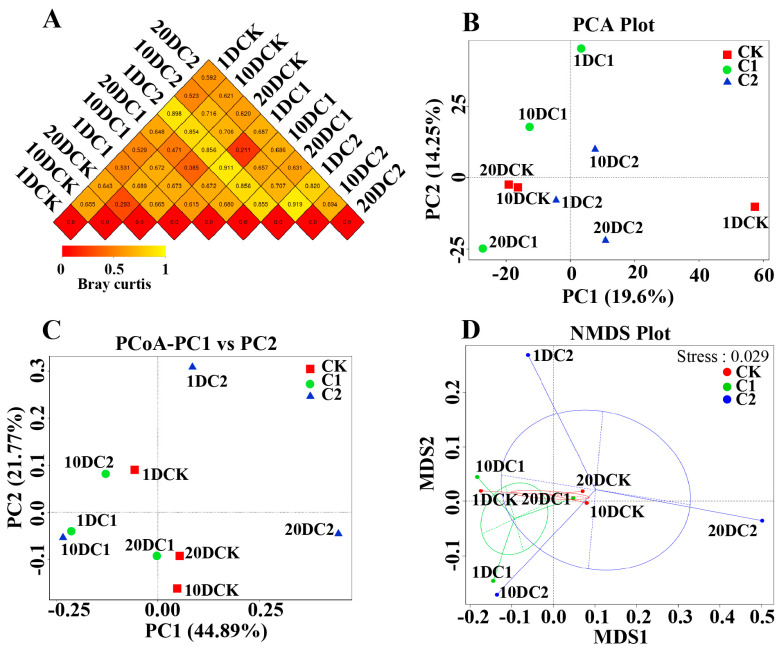
Statistical analysis of bacterial community: (**A**) beta-diversity analysis, (**B**) principal component analysis (PCA), (**C**) principal coordinate analysis (PCoA), and (**D**) Non-metric multidimensional scaling (NMDS) of three different composting treatments. (CK-the blank treatment was static composting with frequently turning; C1-the static composting with microbial inoculants numbered C1; C2-the static composting with microbial inoculants numbered C2; 1DCK-samples in CK collected at first day; 10DCK-samples in CK collected at 10th day; 20DCK-samples in CK collected at 20th day; 1DC1-samples in C1 collected at first day; 10DC1-samples in C1 collected at 10th day; 20DC1-samples in C1 collected at 20th day; 1DC2-samples in C2 collected at first day; 10DC2-samples in C2 collected at 10th day; 20DC2-samples in C2 collected at 20th day).

**Figure 7 microorganisms-11-02605-f007:**
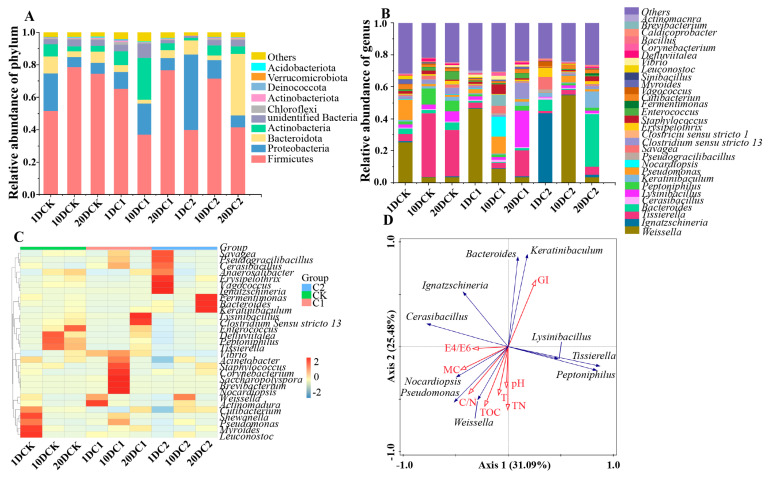
(**A**) Changes in bacterial community at phylum level (top 10). (**B**) Changes in bacterial community at genus level (top 30) during the composting in three treatments. (**C**) Relative abundances of core top 30 genera in different treatments (Colors in the heatmap reflects relative abundances, using red to represent high abundance and blue for low abundance.) (**D**) RDA analysis between environmental factors and the top 10 bacterial community of the three treatments. (CK-the blank treatment was static composting with frequently turning; C1-the static composting with microbial inoculants numbered C1; C2-the static composting with microbial inoculants numbered C2; 1DCK-samples in CK collected at first day; 10DCK-samples in CK collected at 10th day; 20DCK-samples in CK collected at 20th day; 1DC1-samples in C1 collected at first day; 10DC1-samples in C1 collected at 10th day; 20DC1-samples in C1 collected at 20th day; 1DC2-samples in C2 collected at first day; 10DC2-samples in C2 collected at 10th day; 20DC2-samples in C2 collected at 20th day).

**Figure 8 microorganisms-11-02605-f008:**
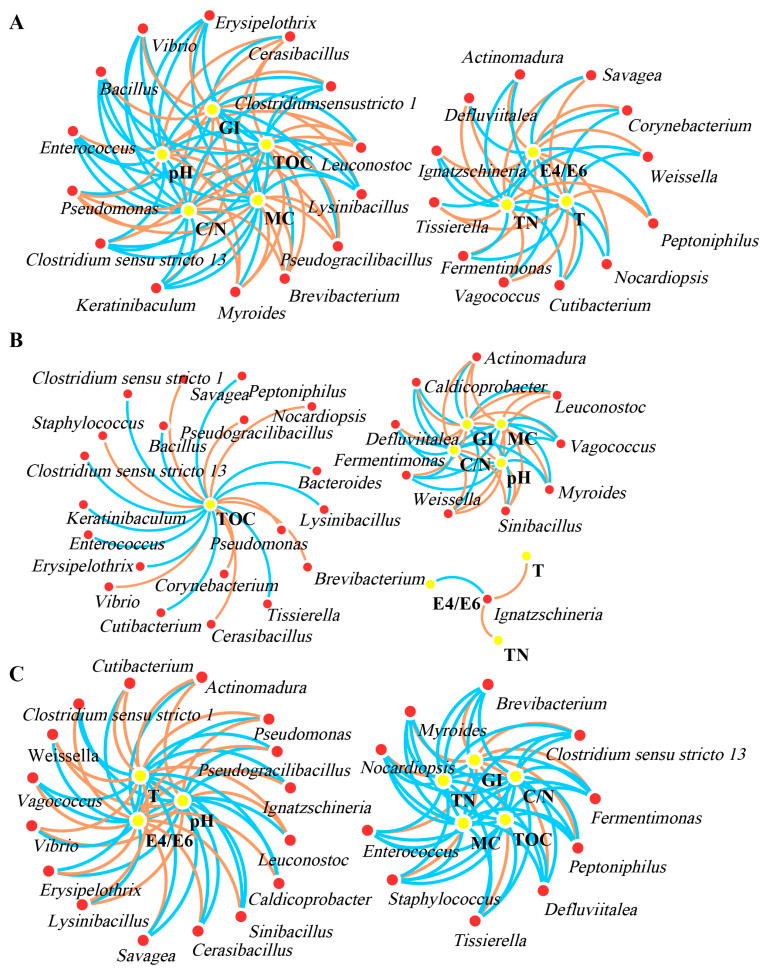
Network analysis between 8 indicators (T, pH, E4/E6, MC, TN, TOC, C/N and GI) and related bacterial genera for CK (**A**), C1 (**B**) and C2 (**C**). Orange arrows displayed the positive correlations among genus and P indicators (*p* < 0.01). Blue arrows indicated the negative correlations (*p* < 0.01). (CK-the blank treatment was static composting with frequently turning; C1-the static composting with microbial inoculants numbered C1; C2-the static composting with microbial inoculants numbered C2).

**Table 1 microorganisms-11-02605-t001:** Microbial richness and diversity in composting samples.

Composting Time (Day)	PD Whole Tree	Chao ^1^	Observed Species	Shannon Index	Simpson Index	Coverage
1DCK	132.102 ± 11.34 a	1435.287 ± 253 a	1307 ± 136 a	6.589 ± 0.92 a	0.956 ± 0.0534 a	0.996 ± 0.0002
10DCK	119.687 + 10.12 b	1257.353 ± 203 a	1140 ± 114 a	5.532 ± 0.89 a	0.907 ± 0.0643 a	0.996 ± 0.0002
20DCK	110.023 ± 11.22 b	1346.273 + 198 b	1229 ± 147 a	6.149 ± 0.84 a	0.945 ± 0.446 a	0.996 ± 0.0004
1DC1	88.633 ± 7.89 b	1199.909 ± 175 b	1095 ± 112 a	6.363 ± 0.76 a	0.961 ± 0.0672 a	0.997 ± 0.0009
10DC1	96.33 ± 6.87 b	1245.359 ± 105 b	1125 ± 89 ab	6.185 ± 1.12 ab	0.873 ± 0.0524 a	0.997 ± 0.0009
20DC1	115.198 ± 10.13 b	1342.944 ± 103 b	1222 ± 94 ab	5.927 ± 1.08 ab	0.939 ± 0.0743 a	0.996 ± 0.0005
1DC2	92.781 ± 6.25 bc	606.038 ± 54 bc	1035 ± 49 bc	4.329 ± 0.61 ab	0.796 ± 0.478 a	0.998 ± 0.0005
10DC2	89.702 ± 5.43 bc	1094.875 ± 104 c	995 ± 67 c	4.678 ± 0.55 b	0.756 ± 0.532 b	0.997 ± 0.0007
20DC2	118.322 ± 7.89 c	1332.793 ± 112 c	1228 ± 117 c	6.111 ± 0.63 b	0.939 ± 0.0714 b	0.997 ± 0.0008

^1^ All values are given as mean ± SD, means follow by the same letter do not differ significantly at *p* < 0.05 level according to the LSD test (*n* = 3). The standard deviation is based on the average of three biological replicates. (1DCK-samples in CK collected at first day; 10DCK-samples in CK collected at 10th day; 20DCK-samples in CK collected at 20th day; 1DC1-samples in C1 collected at first day; 10DC1-samples in C1 collected at 10th day; 20DC1-samples in C1 collected at 20th day; 1DC2-samples in C2 collected at first day; 10DC2-samples in C2 collected at 10th day; 20DC2-samples in C2 collected at 20th day).

## Data Availability

The 16s rRNA sequence data that support the findings of this study are publicly available by the National Center for Biotechnology Information (NCBI, https://www.ncbi.nlm.nih.gov/, accessed on 7 September 2023). The associated BioProject and BioSample numbers are PRJNA1027564 and SAMN37800454, respectively. The raw sequencing data of different samples are obtained through SRR26373766, SRR26373767, SRR26373768, SRR26373769, SRR26373770, SRR26373771, SRR26373772, SRR26373773 and SRR26373774, respectively.

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
