# Peer review of "Effects of Two Different Proportions of Microbial Formulations on Microbial Communities in Kitchen Waste Composting"

_microorganisms, 2023, doi:10.3390/microorganisms11102605_

Round 1

Reviewer 1 Report

Work shows that adding core microbial agents to kitchen waste increased composting efficiency, but time period is short in what cause?. Current paper deals with at least six strains usage, including Firmicutes, Bacteroidota, Actinobacteriota, Proteobacteria, Deinococota, and Chloroflexi for composting. The composing experiments were carried out by selected core bacterial agents and universal bacterial agents for 20 days. The results demonstrated that the addition of core microbial agents effectively controlled the emission of typical odor-producing compounds. The addition of core and universal bacterial agents drastically reduced NH3 emissions by 93.88% and 74.05%, and decreased H2S emissions by 77.74% and 27.01%. The application of core microbial agents during composting elevated the peak temperature to 65℃ and in terms of efficient temperature evolution (>55 °C for 8 consecutive days). The organic matter degradation rate decreased by 24.32% from the initial values for core microbial agents were added, while for the other treatments the reduction was slight- give meaning to “slight”, add results with st deviation.

 Overall graph is needed with overall setup displaying

Methods need better introduction, i.e DO concentrations involved in each phase, SRT, reduction rate, pH, conductivity in different stages.

Experimental setup needs showing sampling vessels, which are enabling fast pollutant removal and change in compost parameters before sampling. Further remarks could be made on MS quality:

1.      R21“Network and RDA analysis”- at first mentioning RDA and other terms need to be defined.

2.      -Data should not interfere with data points error bars (Fig. 2). Some values are not visible, check other Figs also.

3.      Methods. R120 “The E4/E6 ratio was measured using an ultraviolet spectrophotometer at wavelengths 465 nm and 665 nm”- give meaning of this ratio, what it shows.

Chemical oxygen demand (COD), total nitrogen (TN), total phosphorus (TP), calcium ion (Ca2+), SVI, dry weight/ VSS measurement Chinese Standards need showing by ref- give exact methods- Nessler, other colorimetric?

4.      What was exact temperature of process, salinity, reactor DO, PHA and other controlled factors values within phase need defining.

5.      Spacing needed to add after numerical values and before units, check everywhere.

6.      Figures- st. deviation missing, check elsewhere. Fig. 3, 6, 7 and others, enlarge to make them visible.

7.      Nitrogen species (organic N, ammonium, nitrite, nitrate) and COD/BOD ratio, removal rates could be used for showing in addition to % reduction.

8.      Novelty and aims are too vague, specify in more concrete way.

9.      Parallel tests with control need to be done. Influent P, nitrite and nitrate missing as COD, TOC and well their removals

10.   P3.R.116 “NH3 and H2S concentrations in the ambient air were directly measured using a sensor”- is it better to use chromatograph for more exact measurement of emissions.

11.   Proper subscripts need to be included, for legend and elsewhere, all figures should contain proper subscription for chemicals correctly formatted in the text. Also error bars are missing from data points and p values.

12.   P3. R105 “C2 contained six strains, including Firmicutes, Bacteroidota,  Actinobacteriota, Proteobacteria, Deinococota, and Chloroflexi. Microorganisms should be in italics. MS displays well known microorganisms, give comparisons with literature where different results have been shown. Language quality needs significant improvement.

 13.   And the CO2 production should be checked as well together with methane emissions. Figs- use Origin program to generate graphs in better way.

14.   Check ways to estimate anaerobic digestion and pollutants species removal and anoxic treatment possibilities to reduce through autotrophic processes: https://doi.org/10.1080/19443994.2015.1094421, https://doi.org/10.3390/w13212959 https://doi.org/10.3390/w14132063 https://doi.org/10.1080/09593330.2012.665487, https://doi.org/10.1080/09593330.2013.874492

15.   R143 “2.3. Statistical analysis. The data were presented as mean ± standard deviation. Figures were generated using Origin 2021 software”- all this text is not related to Statistics. The residual standard deviations will be need for results justification.

16.   P-values showing between tests seems to be realistic way proving Your observations.

17.   Statistics should be better introduced why certain statistics were chosen, such as „Simpson" etc

18.   Check that if Your calculations are correct and results reliable with available data, with relevant comparisons with literature.

 19. R183 "The degradation of TOC in C2 (28.05%) was the 182 most substantial, whereas there was no apparent decrease in TOC in CK (conventional  composting) and C1 (less than 10% degradation)- not high degradation, elaborate.

20.  R193 "Statistical analysis revealed a significant difference (p=0.008) in C/N values among the four treatments at the end of composting"- comparison with what period was done needs elaborated in all p value cases

ok

Author Response

Dear Editor:

Thanks for your and reviewers’ comments and suggestions on our MS (2612874), which are valuable for the improvement of our paper and later researches. We checked thoroughly and corrected it carefully, and hope the new version is qualified for the publication.

According to your suggestion, we updated the institutional email addresses in the manuscript.

The follows are point-by-point responses to reviewers:

Reviewer #1:

  1. R21“Network and RDA analysis”- at first mentioning RDA and other terms need to be defined.

Answer: Thank you for your careful reading. We have defined RDA in revised MS. (Page 1, line 22)

  1. Data should not interfere with data points error bars (Fig. 2). Some values are not visible, check other Figs also.

Answer: Thank you for your careful reading. We remade Fig. 2 to ensure that data not interfere with data points error bars. At the same time, we also made graphical modifications to Fig.1 and Fig.3. (In revised MS, Page 7, line 203; Page 8, line 220; Page 8, line 242)

  1. R120 “The E4/E6 ratio was measured using an ultraviolet spectrophotometer at wavelengths 465 nm and 665 nm”- give meaning of this ratio, what it shows.

Chemical oxygen demand (COD), total nitrogen (TN), total phosphorus (TP), calcium ion (Ca2+), SVI, dry weight/ VSS measurement Chinese Standards need showing by ref- give exact methods- Nessler, other colorimetric?

Answer: Thank you for your careful reading. We have added why we want to give meaning of E4/E6 (In revised MS, Page 4, line 135 to 137). At the same time, the references for the Chinese standard that you mentioned above was supplied. (In revised MS, Page 4, line 139 to 142). Thank you again for your suggestion.

  1. What was exact temperature of process, salinity, reactor DO, PHA and other controlled factors values within phase need defining.

Answer: Thank you for your valuable comment. The controlled factors values within phase was added. The external temperature during the composting process was around 18 ℃. The initial salt content of kitchen waste was less than 1%, and the DO of the reactor was between 15%~20%. The initial pH value of fresh kitchen waste was 7 (In revised MS, Page 3, line 109 to 112). Thank you again for your suggestion.

  1. Spacing needed to add after numerical values and before units, check everywhere.

Answer: Thank you for your careful reading. We have checked the spacing added after the values and before the units in the entire text. For example, Page 1, line 16 “65 ℃”; Page 4, line 131 “1 h”, etc.

  1. Figures- st. deviation missing, check elsewhere. Fig. 3, 6, 7 and others, enlarge to make them visible.

Answer: Thank you for your careful reading. We have added deviation in Fig. 1, 2, 3. We have modified the layout of Fig. 3, 5, 6, 7 to enlarge the image and used a higher resolution image format.

  1. Nitrogen species (organic N, ammonium, nitrite, nitrate) and COD/BOD ratio, removal rates could be used for showing in addition to % reduction.

Answer: Thank you for your valuable comment. COD/BOD ratio was not measured in our experiment. We used removal rates for showing in addition to % reduction in TOC (In revised MS, Page 6, line 199 to 201).

  1. Novelty and aims are too vague, specify in more concrete way.

Answer: Thank you for your valuable comment. We have supplied and updated the introduction and conclusions to clarify our research purpose and innovation more clearly. (In revised MS, Page 3, line 97 to 100; Page 19, line 539 to 542)

  1. Parallel tests with control need to be done. Influent P, nitrite and nitrate missing as COD, TOC and well their removals.

Answer: Thank you for your careful reading and meaningful comments. Parallel experiments have been done for all tests. “all samples were tested in triplicate” have been added in “2.2.Composting analytical methods” in revised MS ( Page 4, line 142). The initial physical and chemical indicators of T, pH, E4/E6, MC, TN, TOC, C/N, and GI were measured in the manuscript. The relationship between bacteria and the 8 indicators were domenstrated by network and redundancy analysis (RDA). The influent P, nitrite and nitrate have not been measured in this study. We greatly appreciate your suggestion, these indicators will be considered in our future research. We have modified the TOC data description in revised MS, Page 6, line 199 to 201.

  1. R.116 “NH3 and H2S concentrations in the ambient air were directly measured using a sensor”- is it better to use chromatograph for more exact measurement of emissions.

Answer: Thank you for your valuable comment. The composting experiment was conducted in an open environment, and it was difficult to exactly collect the gas. Hence, it was not suitable to use chromatograph for measuring of emissions. We used portable NH3 and H2S sulfide detectors in the experiment. The diagram of the cylindrical fermentors (Figure 1) was added to show the locations of gas collections and temperature measurements (In revised MS, Page 4, line 126).

  1. Proper subscripts need to be included, for legend and elsewhere, all figures should contain proper subscription for chemicals correctly formatted in the text. Also error bars are missing from data points and p values.

Answer: Thank you for your careful reading. We have corrected the subscripts of the chemicals formatted in the text. At the same time, the error bars was added in revised manuscript. (In revised MS, Page 7, line 203; Page 8, line 220; Page 8, line 242)

  1. R105 “C2 contained six strains, including Firmicutes, Bacteroidota, Actinobacteriota, Proteobacteria, Deinococota, and Chloroflexi. Microorganisms should be in italics. MS displays well known microorganisms, give comparisons with literature where different results have been shown. Language quality needs significant improvement.

Answer: Thank you for your careful reading and meaningful comments. We have italicized the microorganisms in revised manuscript. The difference of six strains in the microbial inoculum of C2 with existing relevant references have been compared and discussed in revised MS (Page 18 to 19, line 474 to 515). Language quality was improved.

  1. And the CO2production should be checked as well together with methane emissions. Figs- use Origin program to generate graphs in better way.

Answer: Thank you for your valuable comment. Our experimental objective is to observe the degradation effect of microbial preparations on typical odor pollutants, such as NH3 and H2S. We believe that CO2 production and methane emissions are important greenhouse gases in the composting process.And in our recent research,the bacterial agent C2 has been confirmed to effectively control more than ten types of gases in kitchen waste, including CO2, CH4, C2H6O, CH3CHO, C4H8O2, C4HO, etc (data not published). We have modified the layout of Figures in revised manuscript to enlarge the image and used a higher resolution image format. Thank you again for your suggestion.

  1. Check ways to estimate anaerobic digestion and pollutants species removal and anoxic treatment possibilities to reduce through autotrophic processes:

https://doi.org10.1080/19443994.2015.1094421, https://doi.org/10.3390/w13212959,

https://doi.org/10.3390/w14132063, 

https://doi.org/10.1080/09593330.2012.665487, https://doi.org10.1080/09593330.2013.874492

Answer: Thank you very much for your careful reading and providing valuable references. We have updated the discussion section based on the references.

  1. R143 “3. Statistical analysis. The data were presented as mean ± standard deviation. Figures were generated using Origin 2021 software”- all this text is not related to Statistics. Theresidual standard deviations will be need for results justification.

Answer: Thank you for your careful reading and meaningful comments. We have reorganized “2.3. Statistical analysis”. All standard deviations used for proof of results have been updated in the revised manuscript (In revised MS, Page 6, line 161 ).

  1. P-values showing between tests seems to be realistic way proving Your observations.

Answer: Thank you for your careful reading. The issue you mentioned was corrected (In revised MS, Page 5, line 171 to 172).

  1. Statistics should be better introduced why certain statistics were chosen, such as „Simpson" etc.

Answer: Thank you for your careful reading and meaningful comments. We have reviewed “2.3. Statistical analysis” carefully. “Alpha-diversity, including Chao1, Shannon and Simpson diversity index (H), explained the richness and diversity of bacterial community in all samples using Vegan package in R”(In revised MS, Page 5, line 161 to 164).

  1. Check that if Your calculations are correct and results reliable with available data, with relevant comparisonswith literature.

Answer: Thank you for your valuable comment. We have checked the correctness of the calculations, the reliability of the results, and the availability of data in the revised manuscript. In the discussion section, relevant comparisons were made with the literature. You can view it in revised manuscript (In revised MS, Page 6, line 161 to 175; Page 18 to 19, line 474 to 515).

  1. R183 "The degradation of TOC in C2 (28.05%) was the 182 most substantial, whereas there was no apparent decrease in TOC in CK (conventional composting) and C1 (less than 10% degradation)- not high degradation, elaborate.

Answer: Thank you very much for reminding again. We have adopted your suggestion in 7 and used the removal rate 64.93% instead of showing a reduction of 28.05% (In revised MS, Page 6, line 199 to 201).

  1. R193 "Statistical analysis revealed a significant difference (p=0.008) in C/N values among the four treatments at the end of composting"- comparison with what period was done needs elaborated in all p value cases.

Answer: Thank you for your careful reading and meaningful comments. We have compared the detailed processing times for p-value scenarios. (In revised MS, Page 7, line 211 to 213).

Thanks for your patience. Best wishes!

Sincerely yours,

Hairong Jiang, Yuling Zhang, Ruoqi Cui, Lianhai Ren, Minglu Zhang, Yongjing Wang

Reviewer 2 Report

The study investigated the influence of core microorganisms on the variation in bacterial community assembly during a 20-day kitchen waste composting process using two different inoculants, as determined by high-throughput sequencing. This research is important for a number of researchers working in the area of environmental science. The manuscript was well written and designed, and the authors got interesting results. However the work needs some enhancements before it can be published. I recommend major revision of the manuscript based on the following comments:

The main problem is the methodology. How was the experiment conducted? Samples were taken in triplicate, but were the experiment repeated? Was there one composter or three working in parallel?

Why was this composition of C1 and C2 inoculum chosen?

Is the inoculum used - C1 and C2 available on the market or was it specially prepared for the purposes of the research?

How much of C1 and C2 were added? Were preparations C1 and C2 added once, if so, at what point in the experiment.

In which place were the concentrations of NH3 and H2S tested? How was atmospheric air taken into account during the measurement?

I suggest adding a diagram or photo of the research station indicating the places of sampling and measurement of gas concentration and temperature.

Other comments:

In my opinion, Figures 3, 5, 6 and 7 are illegible and need to be corrected. 

Author Response

Dear Editor:

Thanks for your and reviewers’ comments and suggestions on our MS (2612874), which are valuable for the improvement of our paper and later researches. We checked thoroughly and corrected it carefully, and hope the new version is qualified for the publication.

According to your suggestion, we updated the institutional email addresses in the manuscript.

The follows are point-by-point responses to reviewers:

Reviewer #2:

  1. The main problem is the methodology.How was the experiment conducted? Samples were taken in triplicate, but were the experiment repeated? Was there one composter or three working in parallel? Why was this composition of C1 and C2 inoculum chosen?

Answer: Thank you for your careful reading. We have made detailed supplements and updates to “2.1. Composing experience and sampling” to make it easier for you to understand (In revised MS, Page 3, line 102). The microbial inoculants were provided by the Ecological Environment Laboratory of Beijing Technology and Business University. C1 and C2 inoculants were obtained by screening the same mature compost sample by two classmates. Some of the microorganisms in C2 and C1 are identical. C1 has several additional microorganisms which are common bacteria in deodorization. Therefore, by studying the effects of C1 and C2 on the composting process, we focused on exploring the role of core microorganisms in composting, providing further research for the development of simple microbial agents for efficient deodorization and improving compost maturity.

  1. Is the inoculum used - C1 and C2 available on the market or was it specially prepared for the purposes of the research?

Answer: Thank you very much for reminding again. The microbial inoculants were provided by the Ecological Environment Laboratory of Beijing Technology and Business University. C1 and C2 inoculants were obtained by screening the same mature compost sample by two classmates. Some of the microorganisms in C2 and C1 are identical. C1 has several additional microorganisms which are common bacteria in deodorization. Therefore, by studying the effects of C1 and C2 on the composting process, we focused on exploring the role of core microorganisms in composting, providing further research for the development of simple microbial agents for efficient deodorization and improving compost maturity.

  1. How much of C1 and C2 were added? Were preparations C1 and C2 added once, if so, at what point in the experiment.

Answer: Thank you for your careful reading. We have made detailed supplements and updates to "2.1. Composing experience and sampling". The amount, frequency, and timing of C1 and C2 additions have been supplemented in revised manuscript. On days 1, 8, and 15 of composting, we added 3 L of C1 and C2 microbial inoculum to the corresponding cylindrical fermentor in revised MS (Page 4, line 117 to 118).

  1. In which place were the concentrations of NH3and H2S tested? How was atmospheric air taken into account during the measurement?

Answer: Thank you for your careful reading. We have added Figure 1 showing the sampling and measurement locations of gas concentration and temperature (In revised MS, Page 4, line 126). When we detect gas concentration, we placed the detector in the cylindrical fermentor and close the openings. It can effectively reduce the impact of atmospheric air.

  1. I suggest adding a diagram or photo of the research station indicating the places of sampling and measurement of gas concentration and temperature.

Answer: Thank you for your valuable comment. The schematic diagram and measurement points of the research station were visualized in Figure 1. (In revised MS, Page 4, line 126)

  1. In my opinion, Figures 3, 5, 6 and 7 are illegible and need to be corrected.

Answer:These figures had been revised and made clear in revised manuscript.

Thanks for your patience. Best wishes!

Sincerely yours,

Hairong Jiang, Yuling Zhang, Ruoqi Cui, Lianhai Ren, Minglu Zhang, Yongjing Wang

Round 2

Reviewer 2 Report

The manuscript has been improved.

Author Response

Dear Editor:
Thank you very much for your careful reading and suggestions. They have been very helpful to our manuscript (2612874).

Thanks for your patience. Best wishes!

Sincerely yours,

Hairong Jiang, Yuling Zhang, Ruoqi Cui, Lianhai Ren, Minglu Zhang, Yongjing Wang